# Temperature-Sensitive Modified Bentonite Based on NIPAM for Drilling Fluid: Experimental and Molecular Simulation Studies

**DOI:** 10.3390/molecules28093839

**Published:** 2023-04-30

**Authors:** Yi Pan, Xinyue Zhang, Qianru Zhan, Shuangchun Yang, Yanchao Wang, Jian Guan, Gang Yang, Peng Yang, Zain Ullah Abdul Qayum

**Affiliations:** 1Department of Petroleum and Natural Gas Engineering College, Liaoning Petrochemical University, No. 1, West Section of Dandong Road, Wanghua District, Fushun 113001, China; panyi_bj@126.com (Y.P.); zhagxinyue78@163.com (X.Z.); zqr9702@126.com (Q.Z.); 2Engineering Department of Greatwall Well Drilling Company, China National Petroleum Corporation, Panjin 124000, China; gj_04081@163.com (J.G.); shiyan_1997@163.com (P.Y.); 3Greatwall Well Drilling Company, Sludge Department of China National Petroleum Corporation, Panjin 124000, China; yang_1230411@163.com; 4Department of the Institute of International Education, Liaoning Petrochemical University, Fushun 113001, China; 19841305585@163.com

**Keywords:** bentonite, temperature-sensitive polymer, rheology, molecular simulation, drilling fluid

## Abstract

Bentonite is an important component of drilling fluid, whose quality directly affects the safety and economic benefits of water-based drilling fluid. In order to effectively cope with temperature changes, the development of temperature-sensitive modified bentonite is of great significance. In this study, a temperature-sensitive modified bentonite based on NIPAM with excellent temperature sensitivity was developed through intercalation modification. The temperature-sensitive bentonite (CMC-B-NIPAM) was prepared by grafting N-isopropyl acrylamide (NIPAM) onto the surface of calcium bentonite through the dehydration condensation of silane coupling agent KH570 after the intercalation of sodium Carboxymethyl Cellulose (CMC). The synthesis indexes of CMC-B and CMC-B-NIPAM were optimized by the single-factor method. CMC-B-NIPAM was characterized by XRD and FTIR. The temperature sensitivity, rheology, suspensibility, and expansion capacity of CMC-B-NIPAM dispersion were investigated. The results showed that CMC-B-NIPAM had good temperature sensitivity, and the rheological properties of its dispersion showed characteristics of steady flow and temperature thickening in the range of 40–70 °C. A molecular simulation model was established to observe the microsynthesis mechanism of temperature-sensitive modified bentonite based on NIPAM. The results of this study show that CMC-B-NIPAM drilling fluid has the function of ensuring the stability of drilling fluid flow patterns compared to traditional drilling fluids.

## 1. Introduction

The main body of bentonite is montmorillonite, which is a layered silicate clay mineral. It is named “Fort Benton”, a product produced by the local sodium bentonite in the Rocky Mountains [1]. It is often used in metal forging, chemical preparation, oil and gas production, medical treatment, engineering construction, and other industries because of its excellent suspensibility, water absorption and expansion, plasticity, resistance reduction, and adsorption abilities [2]. Due to the unique structure, outstanding thermal stability, and apparent chemical inertia of bentonite, as well as the function of in-phase replacement, the application range of bentonite is widely discussed by many scholars. In order to avoid the original bentonite, which is vulnerable to its own defects or lack of functionality, and enhance the application space of bentonite, bentonite modification optimization has become the focus of bentonite research. At present, the modification of bentonite is usually conducted by physical intercalation, chemical modification, and other methods to change some of its characteristics. After proper modification, some ions and molecules will exchange into the bentonite interlayer or be reorganized and grafted on the bentonite surface, so that some of the original properties of bentonite can be significantly improved or have new unique properties [3].

The intercalation method uses the unique structure of clay minerals to insert the material into its silicate lamellar structure and destroy its lamellar structure so as to realize the composite on the nanometer scale. It is also a widely used synthesis method. Literature shows that the greater the interlayer spacing, the better the rheological properties of organic clay [4]. Based on sodium bentonite, Yan Zhichao et al. [5] developed various types of intercalated modified bentonite by using polymeric hydroxyl iron ions combined with hexadecyl trimethyl ammonium bromide (CTMAB) and sodium dodecyl sulfate (SDS) as intercalation reagents. The experimental results show that different experimental reagents have significant differences in bentonite modification, among which CTMAB has the best effect. Finally, it is concluded that the adsorption capacity of modified bentonite will be proportional to its interlayer distance, which provides an essential reference value for improving the adsorption of bentonite. In order to solve the problem of formaldehyde pollution in adhesives, Shang Xiaoxian et al. [6] used intercalation modification to improve their adsorption capacity to deal with different pollutants according to the characteristics of bentonite, such as less organic matter content, a simple structure, and easy modification. Chitosan (CS) modified by mPEG was intercalated into bentonite to reduce the amount of free formaldehyde (F) in the adhesive. A new type of formaldehyde removal agent for UF resin adhesive was developed. The results show that the product can significantly reduce the number of free F in urea-formaldehyde resin adhesive, and the strongest clearing F efficiency can reach nearly 40%. In addition, it can also improve the viscosity and strength of the original gum and increase the curing time.

At present, the methods used in organic clay modification are mainly divided into the exchange or adsorption of cations between montmorillonite layers by organic compounds in different environments and surface modification [7] by silane crosslinking agents. A coupling agent is a chemical substance with an amphoteric structure. Some groups in the molecule can react with various functional groups on the surface of silicate to form strong chemical bonds. Another group can react with organic matter through physical adsorption, increasing its affinity and compatibility with organic matter.

Grafting refers to the grafting of a monomer onto bentonite molecules by a covalent bond under the action of the initiator. The surface grafting method is a method of linking polymers to the surface of inorganic nanoparticles by chemical reactions. Various vinyl polymers [8] can be easily grafted onto the surface of inorganic particles by coupling reactions on the surface of inorganic particles with organic groups that can be used for graft polymerization, such as ethyl refining or organic groups containing ethyl refining, or organic groups that can produce free radicals (e.g., -ROH, RNH_2_, -R-O-O-R), typically sufficient hydroxyl groups, accessibility, biodegradability, and reaction compatibility with other molecules [9].

In recent years, thermosensitive polymers have become one of the important research directions in emerging materials, and they have been applied in the petroleum industry, such as oil and gas well drilling, oilfield exploitation, oilfield sewage treatment, and so on. Kamal et al. [10] proposed a new type of high temperature and high shear (HTHS) conditions for the heat-sensitive water-soluble polymer. Due to the presence of thermally sensitive monomers, when the temperature exceeds a certain point, the LCST, a physical network, is formed, leading to an increase in viscosity. To effectively solve the problem of high temperatures and high salt reservoir mining difficulties, BASF [11] developed a temperature-sensitive oil displacement product based on hydrophobically modified polyacrylamide (PAM), which can be used to enhance oil recovery. The viscosity of the product can change with temperature, and the thickening behavior is reversible. In addition, British Petroleum, Chevron, Nalco and Texaco, and other companies (Nalco) jointly developed a new type of temperature-sensitive microgel oil displacement agent, which has been put into use [12]. Some scholars have studied the popular adjustment of oilfield working fluid, temperature-sensitive polymer working principle modeling, and other directions [13]. At present, temperature-sensitive polymer-related products have been put into use in the domestic Jidong oilfield and some oilfields in Indonesia and show good temperature-sensitive performance [14].

In order to address the rheological issues of drilling fluid under low-temperature conditions [15], this article utilizes the intercalation effect of sodium Carboxymethyl Cellulose (CMC) and the temperature sensitivity of N-isopropyl acrylamide (NIPAM). The silane coupling agent KH570 was grafted on the surface of sodium bentonite by dehydration and condensation to add temperature responsibility to it. The modified temperature-sensitive bentonite was optimized and characterized, and its temperature sensitivity, rheology, suspensibility, and expansion capacity were further investigated. It aims to provide a new research direction for the development of bentonite functional diversification to enhance the application range of bentonite and its practical value.

## 2. Results and Discussion

### 2.1. Index Optimization

#### 2.1.1. CMC-B

The static sedimentation stability test can be used to accurately reflect the suspensibility of bentonite. The static sedimentation factor (SF) area between 0.5 and 0.52 can reach the application range of drilling fluid additives; in this range, the smaller the SF value, the better the suspensibility of the solution. The effects of different mass ratios of CMC to bentonite, pH, reaction temperature, and reaction time on the static sedimentation stability of CMC-B dispersion were investigated. The results are shown in Figure 1. As shown in Figure 1a, when the ratio of CMC addition is 20–40%, the static sedimentation factor of bentonite dispersion after intercalation gradually decreases and reaches the lowest value when the ratio of CMC addition is 40%. When the CMC concentration exceeds 40%, the static sedimentation factor of dispersion gradually increases with the increase in CMC concentration. It is speculated that when the reaction begins, a small amount of CMC is displaced in the dispersion and enters the interlayer of bentonite to expand the d-value of bentonite and improve its suspensibility. When the concentration of CMC is greater than 60%, the intercalation reaction reaches saturation, and too much CMC will increase the density of the bottom of the solution, thus affecting the suspensibility of the modified bentonite. After comprehensive consideration, it is determined that the mass ratio of CMC to bentonite is 40%, which is the best reaction amount of CMC.

The pH not only affects the reaction solution but also CMC itself. As shown in Figure 1b, when pH is in the range of 6–8, the static sedimentation factor of CMC-B-NIPAM dispersion gradually decreases with the increase in pH and reaches the minimum when pH = 8. When the pH range exceeds 8, the static factor begins to increase gradually. Presumably, because bentonite is in acidic media, its end face with a positive charge, when a larger volume of organic cations or polar molecules into the interlayer repulsion, hindering the exchange between cations. Meanwhile, CMC is in a semi-sol state in an acidic environment, which is not conducive to the reaction [16]. Under neutral and weak alkaline conditions, there is no such repulsion, so the number of polar molecules entering the bentonite interlayer will increase. However, in the strong alkaline solution, CMC will be degraded, and the intercalation reaction cannot be carried out. Therefore, to maximize the benefits, it is advisable to determine the pH value of the CMC solution to be 8.

The reaction temperature is not only directly related to the energy consumption of experimental synthesis but also to the boiling point of the solvent itself. It can be seen from Figure 1c that with the increase in reaction temperature, the static sedimentation factor of modified bentonite decreases first, then increases, and reaches the lowest point at 60 °C. This is because the cation exchange process can be accelerated by increasing the reaction temperature appropriately, and the hydrogen bond effect between CMC molecular chains will be weakened. This makes it easier to insert into the interlayer of bentonite, increasing the d-value and improving the suspensibility. However, when the temperature is too high, CMC will be degraded, reducing the suspensibility. Therefore, the optimum reaction temperature is 60 °C.

The length of the reaction time and the experimental cost will have a great relationship. In order to reduce the interference of experimental conditions, it is particularly essential to determine a reasonable reaction time. It can be seen from Figure 1d that the static sedimentation factor of CMC-B-NIPAM dispersion synthesized under 4–6 h conditions gradually decreased, indicating that the whole system was undergoing an intercalation reaction. The minimum SF at 6 h indicates that the intercalation reaction in the system has reached the most suitable state. As the reaction continues for more than 6 h, the intercalation reaction of CMC and bentonite has basically reached equilibrium. After comprehensive consideration, it can be concluded that to reduce the reaction energy consumption, 6 h is finally selected as the effective synthesis time of CMC-B-NIPAM.

#### 2.1.2. CMC-B-NIPAM

The effects of NIPAM and KH570-B at different mass ratios on the temperature sensitivity of CMC-B-NIPAM dispersion and the effects of KPS addition, reaction temperature, and reaction time on the rheology of CMC-B-NIPAM dispersion were investigated, respectively. The results are shown in Figure 2. The synthetic effect of temperature-sensitive modified bentonite was evaluated by transmittance and absorbance tests. The lower the transmittance, the higher the absorbance, and the better the polymerization of temperature-sensitive polymer NIPAM, the more fully grafted it is with coupling agent KH570-B and the better the modification effect of modified bentonite CMC-B-NIPAM. As shown in Figure 2a, when the ratio of NIPAM:CMC-B-KH570 is 5–10%, the transmittance and absorbance have no obvious change. When the addition amount of NIPAM increased to 15%, the transmittance and absorbance changed (transmittance decreased and absorbance increased), and when the mass ratio of NIPAM:CMC-B-KH570 was greater than 15%, the change range was relatively small, which indicated that the polymerization of thermosensitive molecules was close to saturation. When the addition of NIPAM increases to 25%, the rotor speed will be affected due to the high consistency of the reaction liquid, which will lead to the deposition of a part of CMC-B-KH570 at the bottom of the reaction device, thereby effectively synthesizing CMC-B-KH570 and poly NIPAM. Therefore, this experiment excludes the addition of NIPAM at 25% and above. Finally, considering both meetings of the CMC-B-NIPAN synthesis and ensuring the reduction of experimental equipment loss, the mass ratio of NIPAM:CMC-B-KH570 was determined to be 15%.

In addition to the transmittance and absorbance used to evaluate the synthesis effect of CMC-B-NIPAM, the change in Apparent Viscosity (AV) in rheology can also be a reasonable verification of the synthesis effect of temperature-sensitive modified bentonite, that is, at the same temperature (higher than the LCST value of CMC-B-NIPAM), the higher the AV at the same concentration, the better the synthesis effect. Usually, the mass ratio of KPS and NIPAM control is 1–15% [17]. When measuring the effect of KPS addition on the rheology of temperature-sensitive modified bentonite, the temperature is set to 50 °C. Under this condition, CMC-B-NIPAM is in a temperature-sensitive response state and can resist “heating and viscosity reduction”, and the better the synthesis effect, the stronger the ability. As shown in Figure 2b, when the concentration of KPS is 6–12%, the apparent viscosity of the temperature-sensitive modified bentonite dispersion gradually decreases with increased KPS addition. When the addition ratio is 12–14%, AV begins to rise, and the polymerization rate of NIPAM is moderate. After 14%, AV fluctuates up and down and is unstable. Significantly, when the addition ratio of KPS exceeds 18%, the viscosity decreases rapidly. It was speculated that at the beginning of the experiment, the addition of KPS was low, and the stability of the KPS was poor. With the increase in KPS, the polymerization reaction of NIPAM becomes more adequate, the temperature-sensitive long chain was gradually increased, and the number of hydrophilic groups began to increase, resulting in a decrease in the viscosity of the dispersion. Viscosity increases and is relatively stable when a certain amount of KPS is added to make full use of temperature-sensitive polymers. When the addition ratio is greater than 16%, NIPAM is forced to aggregate rapidly and at a temperature above its LCST value and cannot be effectively connected to KH570-B. On the premise of maximizing the cost-benefit of synthesis, it is most appropriate to determine the KPS addition value at 14%.

The reaction temperature not only affects the energy consumption of the experiment but is also closely related to the activity of potassium persulfate, which in turn affects the polymerization rate of NIPAM. The polymerization kinetics showed that the decomposition rate of the initiator increased with increasing temperature. And the number of primary free radicals in the polymerization reaction will continue to increase [18], which could prompt the polymerization of NIPAM. When the test temperature is 60 °C, CMC-B-NIPAM is in a temperature-sensitive response state, and each group of samples has the best viscosity-increasing effect. It can be seen from Figure 2c that the viscosity of the dispersion is relatively stable in the range of 40–45 °C. In the range of 50–60 °C, the viscosity of the dispersion gradually decreases, indicating that the temperature-sensitive monomer has been polymerized in this range and that the hydrophilic group on the surface of CMC-B-KH570 has played a role. The viscosity of NIPAM-B dispersion gradually increased in the range of 60–65 °C, indicating that KPS had played a role in this temperature range, and NIPAM began to polymerize. The viscosity of the dispersion in the range of 70–75 °C tended to be stable, indicating that NIPAM completed most of the polymerization. Considering the boiling point of the composite solvent itself (about 67 °C), 65 °C was finally set as the optimal reaction temperature for the CMC-B-NIPAM synthesis experiment.

The experimental reaction time is closely related to the energy consumption of the experimental reaction. If the time is too short, it will affect the effectiveness of KPS, and NIPAM cannot be fully polymerized. If the reaction time is too long, it will increase the synthesis cycle and production costs, so determining the reaction time is essential. It can be seen from Figure 2d that when the time is 5–7 h, the rheology of the dispersion gradually decreases, indicating that the surface of CMC-B-KH570 is only linked to a small amount or even no PNIPAM. The concentration of the solution began to significantly increase after 8–9 h. It was inferred that the oxidation and excitation of KPS activated at this time, and NIPAM reacted effectively with CMC-B-KH570. Although the concentration of dispersion increased slightly from 10 to 11 h, the change was not noticeable. It is speculated that most of the temperature-sensitive modification reactions have been completed. With the extension of the reaction time, the viscosity of the CMC-B-NIPAM solution will not be significantly changed. To maximize the synthesis efficiency and minimize the reaction energy consumption, 9 h was selected as the effective synthesis time of NIPAM-B.

### 2.2. Characterization of CMC-B-NIPAM Composites

#### 2.2.1. XRD

Comparative analysis of bentonite before and after modification base on the formula:(1)$$d=\frac{n\lambda }{2\sin\theta }$$
where d is the interlayer spacing of bentonite; θ is the diffraction angle; λ is the diffraction wavelength (λ = 0.1541 nm); and n is the diffraction order (n = 1).

The change in the d-value and diffraction angle (2θ) of the 001 crystal plane (d001-value) of the bentonite is shown in Figure 3a. It can be seen that the position of 2θ has changed significantly and moved to a lower angle. The position of the X-ray reflection has moved forward, and the diffraction peak after intercalation is more acute. Before intercalation, the d-value of bentonite was 1.1159 nm. For the intercalated bentonite, with the entry of CMC, the d-value increases to 1.4483 nm. Comparing the before and after intercalation, the layer of bentonite increases by 0.3324 nm, which indicates that CMC successfully intercalated into the interlayer of bentonite. It can also be seen from the position of the diffraction peak in the figure that some characteristic peaks of bentonite disappeared after CMC was compounded with bentonite, which indicated that the polymer CMC destroyed part of the layered structure of bentonite in the process of inserting into the interlayer of bentonite. In addition, there are many -OH polar groups between the layers of bentonite. When CMC is mixed with the anhydrous ethanol as the medium and under the action of a strong force, the polar groups in CMC (such as -CH_2_COONa) can form hydrogen bonds with -OH between the layers of bentonite. Under the combined action of the internal driving force (hydrogen bond) and external driving force, the polymer realizes the intercalation process.

#### 2.2.2. FTIR

The existence and change in the groups before and after the reaction was determined based on the characteristic absorption peaks of the functional group in the FTIR spectrum. Figure 3b is the qualitative analysis of the infrared spectrum in the process of bentonite modification.

It can be seen from Figure 3b that in the infrared spectrum of the original bentonite, the region from 920 cm^−1^ to 800 cm^−1^, is the position of the characteristic absorption peak of the octahedral structure group Al-O bond [19]; 1043 cm^−1^ and 940 cm^−1^ are Si-O and Si-O-Si bond stretching vibration absorption peaks, respectively, and 1620 cm^−1^ is -OH bending vibration absorption peak.

After the intercalation reaction between bentonite and CMC, the bending vibration at 803 cm^−1^ was enhanced, and the -OH stretching vibration absorption peak in H_2_O at 1620 cm^−1^ was enhanced and widened. In addition, the stretching vibration absorption peak of sodium CMC appeared near 1400 cm^−1^, and the asymmetric stretching vibration peak of -CH_2_ at 3000 cm^−1^. It can be seen that sodium CMC has successfully entered the interlayer of bentonite, and the -COOH-, -O- bond in sodium CMC has coordinated reaction with the -OH, Si-O bond in bentonite, resulting in a new structure. After coupling with KH570, the vibration peak of the -CH_2_ or -CH_3_ group at 3000 cm^−1^ is enhanced, which is due to the presence of organic components in the silane coupling agent KH570, but the effect of physical adsorption is not excluded. It should be noted that the characteristic absorption peak at 1620 cm^−1^ is enhanced. It can be seen that the layer, inter-surface water, and -OH on the surface of bentonite are involved in the hydrolysis, dehydration, and condensation between silane coupling agent KH570 and bentonite. After the temperature-sensitive modification, it can be clearly seen that the characteristic absorption peak of -CH(CH_3_)_2_ is added at 1300 cm^−1^, and the characteristic absorption peaks of the “N-H” bond in the amide II absorption band and the “C-N” bond in the amide II absorption band are added at 1527 cm^−1^ and 1617 cm^−1^, respectively, and the vibration absorption peak at 2000 cm^−1^ is also significantly widened. After the preparation is completed, the PNIPAM residue is retained due to repeated washing and suction filtration of the synthesized product.

### 2.3. Drilling Fluid Performance

#### 2.3.1. Performance Evaluation

(1)Thermosensitive

Testing the transmittance and absorbance of the sample at different temperatures is the easiest and most intuitive way to determine the temperature-sensitive properties of the polymers. Here, 1 g bentonite original (200 mesh) was dispersed in 100 mL distilled water, stirred for 30 min, and then tested. The data are shown in Table 1. The transmittance and absorbance of the bentonite solution were maintained at 0.2 and 2.6, respectively, during the heating process. This shows that temperature changes will not affect the original bentonite transmittance and absorbance of the numerical changes.

In order to verify the temperature-sensitive properties of the modified bentonite and accurately detect the LCST value corresponding to CMC-B-NIPAM, the modified bentonite solution was also tested. The observed concentration was 1 g/mL and tested at room temperature (20 °C). As shown in Table 2, the transmittance and absorbance data of the solution changed at a temperature of 50 °C, in sharp contrast to the recorded data of the original bentonite, and it was inferred that the LCST value range of CMC-B-NIPAM was 45–50 °C.

In order to further determine the LCST value, the temperature test unit interval value is set to 1 °C while ensuring that the instrument can achieve effective recognition and the test data meet the requirements of reducing the experimental error. As shown in Table 3, the temperature at which the transmittance and absorbance data first change is 48 °C. Therefore, 48 °C is determined as the LCST value of CMC-B-NIPAM.

(2)Rheological property

It can be seen from Figure 4a that AV and PV of bentonite decrease with the increase in temperature, especially V, which decreased by 42% from 20 °C to 70 °C. The AV and PV values of bentonite dispersion after 45 °C are significantly lower. Yb also showed a downward trend. Therefore, the rheological properties of bentonite before modification decreased with the increase in temperature-heating viscosity reduction.

The rheological test of bentonite after temperature-sensitive modification was carried out by the same method. It can be seen from Figure 4b that the viscosity of CMC-B-NIPAM slurry is larger than that of unmodified slurry. The rheological properties of CMC-B-NIPAM dispersion decreased significantly at 45 °C, and AV and PV began to show an upward trend. Especially in the range of 50–55 °C, the slope of AV and PV reached the maximum and the increasing trend was the most obvious, which indicated that the effect of temperature-sensitive response in this range was the best. It can be seen that the modified bentonite has a significant temperature sensitivity and a certain temperature-response ability for temperature-sensitive thickening.

(3)Suspensibility

The intuitive test method verifies that the change in bentonite before and after modification is simple and clear. As shown in Figure 5a, a is the original bentonite, b is CMC-B-NIPAM. From the observation results of the two dispersions at 0 h, 6 h, 12 h, and 24 h, it can be seen that in the initial state, both are evenly distributed. The change in solution is not obvious at 6 h, a2 has a slight settlement, a3 settlement is slightly higher than b3 at 12 h, and a4 settlement is significantly higher than b4 at 24 h.

The method of using a static factor to test the suspensibility of bentonite is simple and quick, which makes it suitable for field operations. The SF value as a static settlement factor needs to be within a reasonable range to meet the requirements. When the value of SF is less than or equal to 0.5, it is proven that no static settlement occurs at this time. When the SF value is higher than 0.52, it indicates that the settlement is too apparent, the static stability is insufficient, and the requirement is not met. It can be seen from the calculation results in Table 4 that the SF values of both are less than 0.52. The SFBent value is 0.5144, and the SFCMC-B-NIPAM is 0.5023. It can be seen from the calculation results that the SF values of bentonite and CMC-B-NIPAM are lower than 0.52, indicating that the suspensibility of both is relatively good and meets the requirements. However, the dispersion capacity of bentonite after temperature-sensitive modification is slightly stronger than that of bentonite, which can better meet the application requirements.

(4)Expansion capacity

As one of the technical indexes for evaluating bentonite quality, swelling capacity is also an essential reference for judging the quality of bentonite. The ability of interlayer adsorption and hybridization of bentonite is an important manifestation of its particle size. The size of the swelling capacity is proportional to the adsorption performance and excellent quality of bentonite. Therefore, it is necessary to test the swelling capacity of bentonite before and after modification.

As shown in Figure 5b, the state diagrams of static settlement volume at different time points are shown, respectively. Among them, a is bentonite dispersion, and b is temperature-sensitive modified bentonite dispersion; a1 and b1 are the solutions in the initial placement state. It can be seen that the dispersion of bentonite in the two solutions is more uniform, a small amount of floc is formed, and the settlement is not apparent. Here, a2 and b2 are the solutions after standing for 12 h. It can be seen from the diagram that there are apparent differences between the two sample dispersions of a2 and b2 at this time. Both solutions have precipitates, but solution a2 has apparent stratification and the upper part is transparent. It can be seen that both a3 and b3 solutions have entirely settled, and the upper part of the solution is completely transparent. Moreover, due to the effect of hydrochloric acid, the bentonite sediment after standing for 24 h presents a flocculent state rather than simple bentonite sediment. Through observation and calculation, the VS of bentonite is 3 mL/g, and the V_S_ of intelligent temperature-sensitive bentonite is 6 mL/g. It can be seen that the expansion capacity of CMC-B-NIAPM is slightly better than that of bentonite.

#### 2.3.2. Comparison Test

“400 mL distilled water + 4% CMC-B-NIPAM + 0.5% NaCl + 0.2% limestone + 0.2% polyanionic cellulose + 3% polyacrylamide” was used as a temperature-sensitive modified drilling fluid formulation, and “400 mL distilled water + 4% drilling special bentonite + 0.5% NaCl + 0.2% limestone + 0.2% polyanionic cellulose + 3% polyacrylamide” was used as a conventional drilling fluid control group formulation.

In the range of 20–70 °C, the rheology of drilling fluid is tested with 5 °C as the unit threshold. The relevant data are shown in Figure 6a,b, which are the rheological data (mainly AV and PV) of conventional drilling fluid and temperature-sensitive modified drilling fluid at different temperatures. It can be seen from Figure 6a that the AV and PV of conventional drilling fluid decrease with an increase in temperature. AV decreased by nearly 45%, while PV decreased by 53%, especially in the range of 55–60 °C. This is because the active bentonite particles in the drilling fluid increase rapidly due to the influence of temperature rise, and the mutual movement between solid particles becomes easier, resulting in a change in the rheology of the drilling fluid [20]. The temperature-sensitive modified drilling fluid in Figure 6b shows different states. The AV and PV values decrease with the increase in temperature in the range of 15–45 °C and change after 45 °C, that is, the values of the two increase significantly in the range of 45–60 °C, and remain stable after 60 °C. This is because when the temperature reaches 45 °C, the temperature-sensitive modified bentonite reaches the LCST value. At this time, the long chain of poly NIPAM of the temperature-sensitive modified bentonite begins to stretch and play its role, so that the rheology of the drilling fluid begins to increase. After 60 °C, the long chain of poly NIPAM is fully stretched, resulting in the rheology of the drilling fluid being stable. Therefore, the increase in temperature-sensitive viscosity of temperature-sensitive modified drilling fluid is verified. Compared with the drilling fluid prepared by special bentonite for drilling, it can ensure the flow pattern stability of the drilling fluid.

### 2.4. Molecular Simulation

#### 2.4.1. CMC-B

CMC (structural formula shown in Figure 7a) is a common anionic organic polymer containing polyhydroxy and carboxyl groups. Its molecular formula is usually [C_6_H_7_O_2_(OH) _2_OCH_2_COONa]. Among them, the polar groups in CMC (such as -CH_2_COONa, etc.) are easy to break and enter the bentonite layer under the combined action of the internal driving force (hydrogen bond) and the external driving force. In addition, the -O- and -H- bonds in CMC are also easily damaged and fractured. CMC is cheap and easy to obtain. It is often used in industry as a gel-forming agent and an improver of aqueous or dispersion fluidity [21]. As a modifier, it can be intercalated into bentonite to improve its suspensibility. The chemical properties of bentonite are very complex, and it is easy to compensate for the charge balance by cations with large interlayer adsorption. In addition, the Si-O bond and Al-O bond on the edge of bentonite are easy to break, resulting in a positive or negative charge of bentonite. So that it can carry out ion exchange. After ion exchange, the d-value of bentonite will increase, and the suspensibility will also be improved.

As shown in Figure 8a, it is a simple structure diagram of bentonite. Figure 8b is the structure diagram of CMC-B, where the red is O, the white is H, the purple is Al, the green is Ca, the yellow is Si, the coffee is Mg, and the blue is Na. It can be clearly seen from the figure that Ca^2+^ does not appear in Figure 8b, but a large amount of Na+ appears in the modified bentonite. It can be seen that Na+ has successfully entered the interlayer of bentonite, expanding the interlayer structure of CMC-B and forming a relatively stable d-value. Therefore, CMC completed the intercalation process and realized the intercalation modification of bentonite.

#### 2.4.2. CMC-B-KH570

Silane coupling agent KH570 (the structural formula is shown in Figure 7b), as one of many silane coupling products, is a kind of small molecule silicone compound with a unique structure. Its structural formula can usually be simplified by RSiX3, where R can be expressed as an epoxy group, a vinyl group, an amino group, or other organic functional groups that are easy to react with bentonite or exist on the surface of bentonite by adsorption. After being dissolved in water, KH570 will first react with water [22] and exist in the aqueous solution in the form of silanol. The increase in temperature will cause the hydrolysis of silanol, which will then react with the -OH on the surface of bentonite to form hydrogen bonds. Then, dehydrate and retract again to form “-SiO-”, and the silanol will interact with each other and cover the surface of bentonite in the form of a network structure [23].

The formed “-SiO-” will bond with Al-OH and Si-OH on the surface of bentonite. At this time, the electronegativity of the two atoms of hydrogen and oxygen is quite different, causing the electron group to shift rapidly in the direction of the oxygen atom, resulting in the uneven electrical arrangement of the hydrogen atom. In this case, the hydrogen atom easily forms a new hydrogen bond with the silane. The temperature rises again causing the original structure of H_2_O to emerge again and generate a covalent structure, and ultimately a wide range of silane coupling agents on the surface of bentonite modification. The surface-modified bentonite (CMC-B-KH570) structure is shown in Figure 9a, a large number of silane coupling agents grafted onto the surface of bentonite to complete the surface pretreatment of bentonite, through further observation and analysis of bentonite can be effectively connected with KH570 “-O-” and the structure is relatively stable. At the same time, it also provides a “-C=CH_2_” bond for subsequent temperature-sensitive modification.

#### 2.4.3. CMC-B-NIPAM

NIPAM (the structural formula is shown in Figure 7c) is an organic compound containing both hydrophobic isopropyl and hydrophilic amide groups because the hydrophilic and hydrophobic groups contained in it can exhibit different hydrophilicity and hydrophobicity under different temperature stimuli [24]. As shown in Figure 7d, the NIPAM monomer is polymerized into PNIPAM. While the NIPAM monomer is polymerized to form a temperature-sensitive long chain, it will also react with the “-C=CH_2_” bond on KH570-B to complete the effective connection between the PNIPAM temperature-sensitive long chain and bentonite to achieve the purpose of intelligent temperature-sensitive modification of bentonite, as shown in Figure 9b.

When the temperature is lower than LCST, the hydrophilic group plays an important role, and the temperature-sensitive long chain of NIPAM will dissolve in the aqueous solution. Under the interaction of water molecules and hydrogen bonds, a layer of outer hydration will be formed near the temperature-sensitive long chain. The hydration outer layer structure is orderly and relatively stable, which promotes the temperature-sensitive long chain to exhibit a stretchable long-line state. When the temperature is higher than the LCST value, the increased temperature causes the energy of the whole system to increase, resulting in the enhancement of the activity of water molecules and the intensification of thermal motion, which causes the hydration outer layer formed near the temperature-sensitive long chain to begin to rupture or even disappear. At the same time, due to the increase in temperature, the activity of hydrophobic groups on the long chain of high temperature-sensitive molecules will also increase, and the hydrophobic effect will be enhanced. Under this action, a layer of hydrophobic film will be formed. The formed film is relatively compact, and the similar temperature-sensitive long chain is also tightly compressed, thus showing a dense cluster structure and even reaching a dense colloidal particle state.

In other words, CMC-B-NIPAM will be in a free state at room temperature (lower than the LCST value), and the long chain of temperature-sensitive molecules formed on the surface will show a stretch state. At this time, the interaction between temperature-sensitive modified bentonite monomers will be reduced, the mobility will be enhanced, and the interaction between monomers will be weakened. When the temperature rises until it exceeds the LCST of CMC-B-NIPAM, the hydrophobic effect of the hydrophobic group is enhanced, and the long chain of the temperature-sensitive molecule will contract violently so that the monomers are entangled with each other. The interaction between the modified bentonite particles is constrained, so the viscosity of the bentonite changes, resulting in the temperature-sensitive modified bentonite dispersion. After being affected by temperature, the effect of “heating and viscosity reduction” is achieved, and the purpose of rheological self-regulation is achieved.

## 3. Experiment

### 3.1. Materials

Ca-based bentonite was provided by Jianping Jiaxin Chemical Co., Ltd. (Chaoyang, China). Sodium carboxymethyl cellulose (CMC, purity ≥ 98%), silane coupling agent (KH570, purity ≥ 98%), N-isopropyl acrylamide (NIPAM, purity ≥ 98%), potassium persulfate (KPS, purity ≥ 99.5%), tetrahydrofuran (H_2_O/THF, purity ≥ 99.5%), and anhydrous ethanol (purity ≥ 99.7%) were used.

### 3.2. Preparation of Modified Bentonite

#### 3.2.1. Preparation of Intercalated Bentonite (CMC-B)

The bentonite powder (Ca-based) was dried in a vacuum drying oven at 120 °C for 24 h. A certain amount of bentonite was weighed and dispersed in 50 mL anhydrous ethanol, ultrasonically dispersed for 30 min, and then transferred to a three-necked flask. The pH of sodium carboxymethyl cellulose was adjusted with 0.1 mol/L HCL or NaOH [25], and the adjusted sodium carboxymethyl cellulose solution was poured into the three-necked flask and placed in a supergene oscillator to adjust the reaction temperature and react for a period of time. Then, the product was centrifuged, washed several times with anhydrous ethanol, and dried in a vacuum oven at 120 °C for 24 h, named “CMC-B”.

#### 3.2.2. Surface Modification Body Preparation (CMC-B-KH570)

A certain amount of CMC-B and silane coupling agent KH570 were dissolved in 90 mL and 10 mL anhydrous ethanol, respectively, and ultrasonically dispersed for 30 min. The bentonite alcohol and alcohol dispersions containing silane coupling agent KH570 were transferred into a three-necked flask and stirred at 70 °C for 6 h. After the reaction, the product was centrifuged, washed several times with anhydrous ethanol, and dried in a vacuum drying oven at 60 °C for later use, named “CMC-B-KH570”.

#### 3.2.3. Synthesis of Temperature-Sensitive Modified Bentonite Based on NIPAM (CMC-B-NIPAM)

A certain amount of N-isopropyl acrylamide was dissolved in a mixed solvent of H_2_O/THF (tetrahydrofuran) with a volume ratio of 2:1, and CMC-B-KH570 was added to ultrasonically disperse for 30 min. Then, a certain amount of potassium persulfate was added and stirred quickly to make the system uniform; then, it reacted at 65 °C under N_2_ environmental protection for 9 h. After the reaction, the product was centrifuged, washed several times with anhydrous ethanol, and dried in a vacuum drying oven at 60 °C, named “CMC-B-NIPAM”. The preparation of temperature-sensitive modified bentonite based on NIPAM was completed.

### 3.3. Methods

#### 3.3.1. Indicator Optimization

The synthesis process of temperature-sensitive modified bentonite has complex steps [26], which are easily affected by many experimental factors. In the process of intercalation modification, factors such as the mass ratio of CMC to bentonite (20%, 40%, 60%, 80%, 100%), pH (6, 7, 8, 9, 10), reaction temperature (40° C, 50 °C, 60 °C, 70 °C, 80 °C), and reaction time (4 h, 5 h, 6 h, 7 h, 8 h) were investigated. In the synthesis process, the mass ratio of NIPAM to KH570-B (5%, 10%, 15%, 20%, 25%), KPS addition value (6%, 8%, 10%, 12%, 14%, 16%, 18%, 20%), reaction temperature (40 °C, 45 °C, 50 °C, 55 °C, 60 °C, 65 °C, 70 °C, 75 °C), and reaction time (5 h, 6 h, 7 h, 8 h, 9 h, 10 h, 11 h) were optimized.

#### 3.3.2. Microscopic Characterization

The bentonite before and after modification was characterized by X-ray diffraction (XRD) and Fourier transform infrared spectroscopy (FTIR), respectively, to more accurately verify the reliability of the experiment. XRD, as the most basic and extensive structural testing method in material analysis, can not only accurately complete phase analysis but also accurately measure the microcrystalline volume, crystallization status, and crystal orientation of the objects. FITR can analyze the frequency positions of absorption bands or obtained bands generated by molecules that absorb infrared light, and complete the chemical composition and molecular architecture determination of materials. This article used XRD to test the original soil, CMC-B, CMC-B-KH570, and CMC-B-NIPAM and calculated the spacing between d001 crystal planes through XRD. FTIR was used to characterize calcium-based bentonite, CMC-B, CMC-B-KH570, and CMC-B-NIPAM, in order to accurately determine the synthesis effect of CMC-B-NIPAM and gain a more accurate understanding of the structural composition of CMC-B-NIPAM.

#### 3.3.3. Drilling Fluid Properties

(1)Transmittance and absorbance

An appropriate amount of bentonite and CMC-B-NIPAM were dispersed in distilled water, thoroughly stirred, and heated. When the sample was heated to the different required temperatures, it was put into a cuvette [23] and placed in the ultraviolet spectrophotometer. The corresponding readings were recorded at the 5th second, and the above steps were repeated. The absorbance and transmittance data of the samples at different temperatures were recorded, and the average value was taken three times; that is, the absorbance and transmittance tests of the two bentonites were completed. Observing the two test values, when the data changed significantly, it was the LCST value of the solution. The ultraviolet-visible spectrophotometry completed the judgment of temperature sensitivity and the determination of the LCST value of the solution.

(2)Rheological property

The rheological properties of temperature-sensitive modified bentonite CMC-B-NIPAM were tested according to the national standard (GB/T 5005–2010, drilling fluid material specification). Bentonite and CMC-B-NIPAM were weighed individually at 22.5 g, then dispersed in 350 mL distilled water and stirred under a high-speed mixer for 20 min. If too much bentonite adhered to the inner wall of the sample cup, it was stirred for 5 min ± 0.5 min after removing the container from the stirring machine and scraping the adhesion in the cup wall bentonite. The bentonite dispersion was sealed and cured at room temperature for 16 h after complete mixing, and the curing temperature was recorded. After the curing was complete, the liquid was evenly poured into the mixing container and stirred for 5 min before the measurement. Then, it was placed into the sample cup of the six-speed rotary viscometer to prepare for viscosity measurement. The sample was placed in a cup on the tray, adjusting the height of the tray so that the height of the outer cylinder scale line is consistent with the height of the sample liquid level, and then fixing the tray. During the measurement, the rotation speed went from high to low. After the dashboard pointer was stable, the reading was carried out and the data were recorded. Then, the rheology of bentonite dispersion was calculated using the following formula:(2)$$\mathrm{AV}=\theta_{600}/2$$
(3)$$\mathrm{PV}=\theta_{600}-\theta_{300}$$
where $\theta_{600}$ corresponds to the scale value of the viscometer when the drilling speed is 600 r/min. $\theta_{300}$ corresponds to the scale value of the viscometer when the drilling speed is 300 r/min. AV is apparent viscosity. PV is plastic viscosity.

(3)Suspensibility

Two methods were mainly used to test suspensibility: the direct observation method and the static settlement stability method. The direct observation method is as follows: 6.24 g of bentonite and modified bentonite CMC-B-NIPAM were taken and dispersed in 100 mL distilled water. The speed of the high-frequency stirrer was set at 11,000 r/min and was stirred at this speed for 5 min. Then, the solution was placed in a measuring cylinder to be plugged, and the settlement changes of 0 h, 6 h, 12 h, and 24 h in the measuring cylinder were observed, respectively.

In the static settlement stability test, 22.5 g of each sample bentonite was evenly dispersed in 350 mL distilled water and began to stir with a frequency conversion high-speed mixer. After 5 min, the container was removed, the bentonite adhered to the stirring sample cup down was scrapped, and then the container was placed in the stirrer until the end of stirring after 20 min. Subsequently, the two dispersions were poured into a prepared stainless steel tank and placed at room temperature for 24 h. After settling, the two sample solutions were poured into the YM-3 liquid densimeter to measure the upper density ($\rho_{\mathrm{top}}$) and the bottom density ($\rho_{\mathrm{botton}}$), and the measurement data were recorded.
(4)$$\mathrm{SF}=\rho_{\mathrm{bottom}}/\left(\rho_{\mathrm{top}}+\rho_{\mathrm{bottom}}\right)$$

SF is the static settlement factor. $\rho_{\mathrm{top}}$ is the density of the upper portion of the column (lower portion of the free liquid), g/cm^3^. $\rho_{\mathrm{bottom}}$ is the density at the bottom of the column, g/cm^3^.

(4)Expansion capacity

First, for the deployment of the hydrochloric acid solution, the measured concentration of 1 mol/L of 83 mL hydrochloric acid solution was taken, and water was added to dilute the solution to 1000 mL for later use. The bentonite was dried, then 1 g was weighed for later use and prepared with a plug measuring cylinder (100 mL ± 0.5 mL). Then, 50 mL of distilled water was weighed into the measuring cylinder with a stopper, bentonite was placed in it, the stopper of the measuring cylinder was tightened, and the measuring cylinder was shaken at a constant speed for 150 groups. After the bentonite was effectively dispersed in the measuring cylinder, 25 mL of the prepared hydrochloric acid solution was added to the measuring cylinder, and distilled water was added until it flushed with the 100 mL scale line, the stopper was sealed, and 100 groups were continuously shaken. The measuring cylinder with a stopper was placed on a stable table in static settlement for 24 h, and the deposition data were observed and recorded.
(5)$$V_{S}=v/m$$

$V_{S}$ is the expansion volume, mL/g. v is the observed settlement volume of the sediment, mL. m is the amount of sample added.

## 4. Conclusions

This article mainly investigates the preparation and performance of temperature-sensitive modified bentonite CMC-B-NIPAM. After single factor experiment, the optimum synthesis conditions of CMC-B-NIPAM were determined as follows: the mass ratio of NIPAM to KH570-B is 15%, the mass ratio of KPS to NIPAM is 14%, the reaction temperature is 65 °C, and the reaction time is 9 h. Through the multivariate microscopic characterization of XRD and Fourier-infrared spectroscopy, it can be seen that CMC successfully intercalated into the interlayer of bentonite, and expanded the d-value. At the same time, NIPAM is effectively bonded to the surface of CMC-B-KH570 through the coupling effect of KH570, thus completing the temperature-sensitive modification of CMC-B-NIPAM. The LCST value of CMC-B-NIPAM was determined to be 48 °C by measuring the transmittance and absorbance of the solution.

The apparent viscosity and plastic viscosity of ordinary bentonite slurry solution show a decreasing trend at 20–70 °C, while the temperature-sensitive modified bentonite base slurry solution shows a decreasing trend of AV and PV before 45 °C, but shows an increasing trend of AV and PV in the range of 40–70 °C, especially in the range of 50–55 °C. These results demonstrate the temperature sensitivity of modified bentonite and the enhancement of the rheological properties of bentonite slurry under low-temperature conditions. By measuring the SF value, it can be determined that the suspension of the modified bentonite slurry is slightly stronger than that of the original bentonite. Finally, the superiority of modified bentonite in low-temperature rheological properties was further verified by examining the performance of drilling fluid prepared with modified bentonite. The present study also simulates the process of modifying bentonite from a microscopic perspective, in order to directly observe and study the structure and properties of modified bentonite under microscopic conditions.

## Figures and Tables

**Figure 1 molecules-28-03839-f001:**
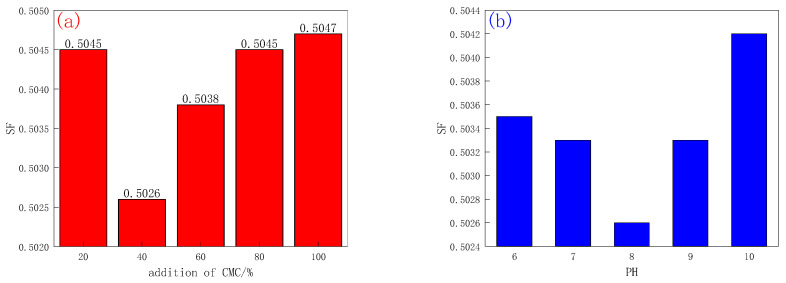
(**a**) The effect of CMC addition on the SF value of CMC-B dispersion, (**b**) The effect of pH on the SF value of CMC-B dispersion, (**c**) The effect of reaction temperature on the SF value of CMC-B dispersion, and (**d**) The effect of reaction time on the SF value of CMC-B dispersion.

**Figure 2 molecules-28-03839-f002:**
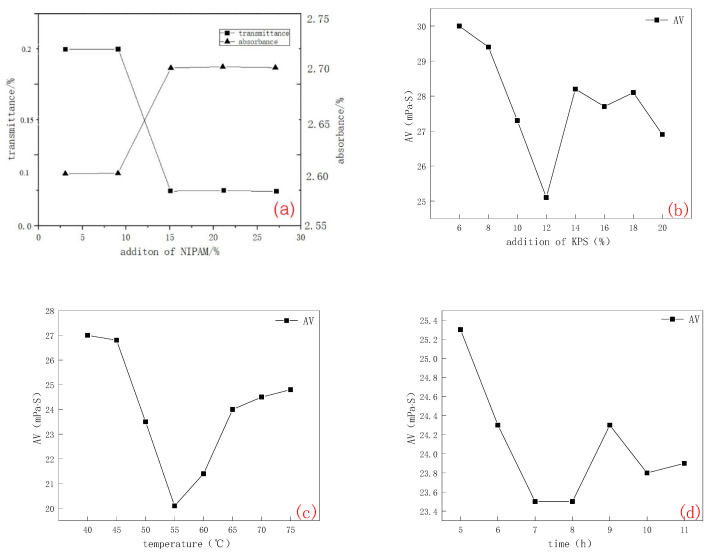
(**a**) The influence of NIPAM on the transmittance and absorbance of CMC-B-NIPAM dispersion, (**b**) The effect of KPS addition on the rheology of CMC-B-NIPAM dispersion, (**c**) The effect of reaction temperature on the rheology of CMC-B-NIPAM dispersion, and (**d**) The effect of reaction time on the rheology of CMC-B-NIPAM dispersion.

**Figure 3 molecules-28-03839-f003:**
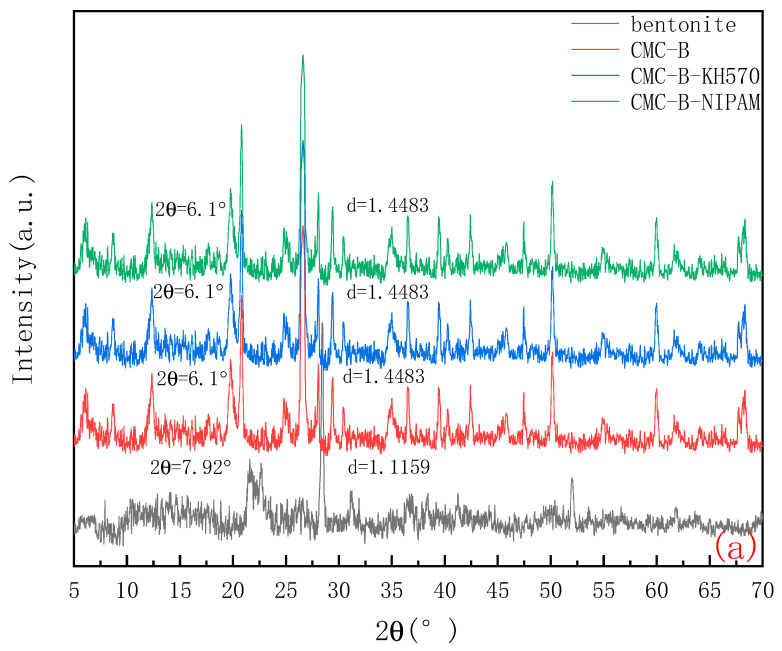
(**a**) X-ray diffraction spectrum and (**b**) Fourier-infrared spectrum.

**Figure 4 molecules-28-03839-f004:**
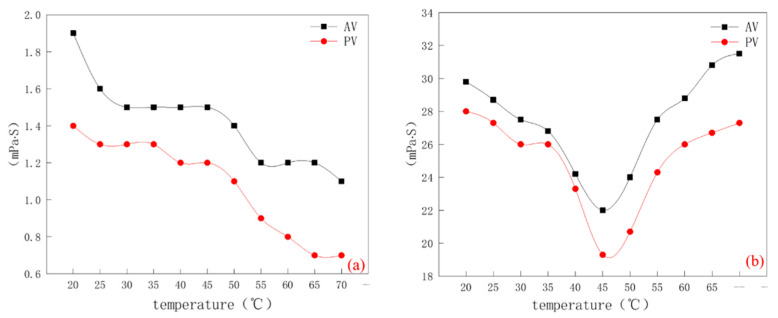
(**a**) Rheological fluctuation diagram of bentonite at different temperatures and (**b**) Rheological fluctuation diagram of temperature-sensitive modified bentonite at different temperatures.

**Figure 5 molecules-28-03839-f005:**
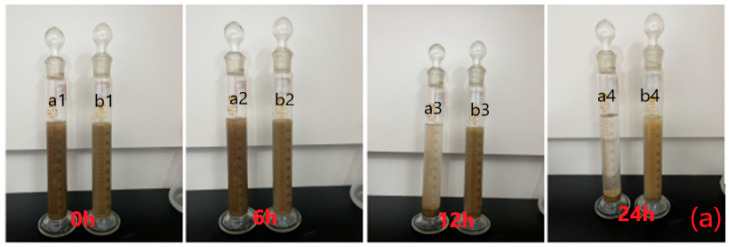
(**a**) Dispersion test of temperature-sensitive modified bentonite and (**b**) Expansion capacity test of intelligent temperature-sensitive modified bentonite.

**Figure 6 molecules-28-03839-f006:**
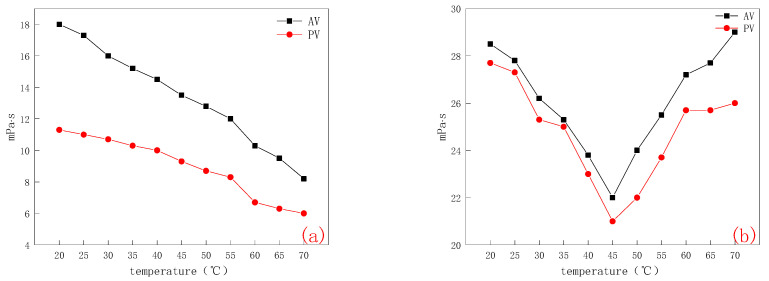
(**a**) Rheological fluctuation diagram of conventional drilling fluid at different temperatures and (**b**) Rheological fluctuation diagram of temperature-sensitive modified drilling fluid at different temperatures.

**Figure 7 molecules-28-03839-f007:**
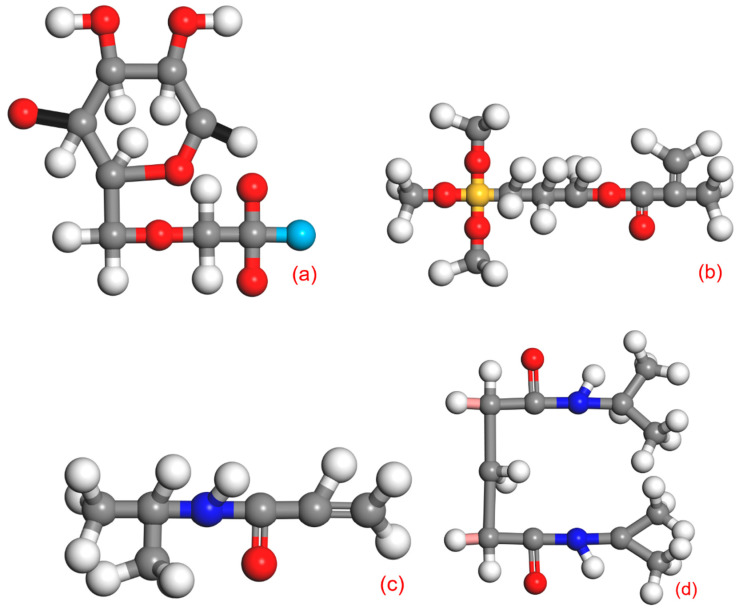
(**a**) Schematic diagram of the molecular structure of CMC, (**b**) Schematic diagram of the molecular structure of silane coupling agent KH570, (**c**) Structure diagram of N-isopropyl acrylamide monomer, and (**d**) Structure diagram of N-isopropyl acrylamide monomer polymerization.

**Figure 8 molecules-28-03839-f008:**
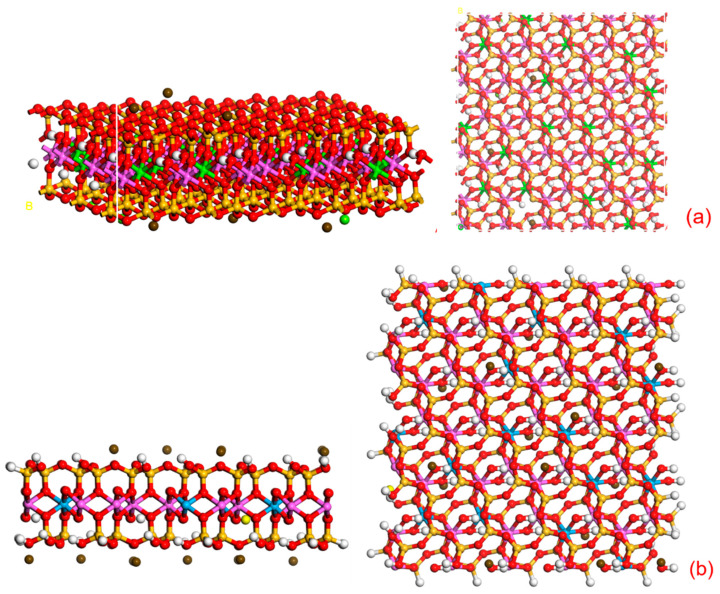
(**a**) Schematic diagram of bentonite structure and (**b**) Schematic diagram of the structure of bentonite (CMC-B) after intercalation.

**Figure 9 molecules-28-03839-f009:**
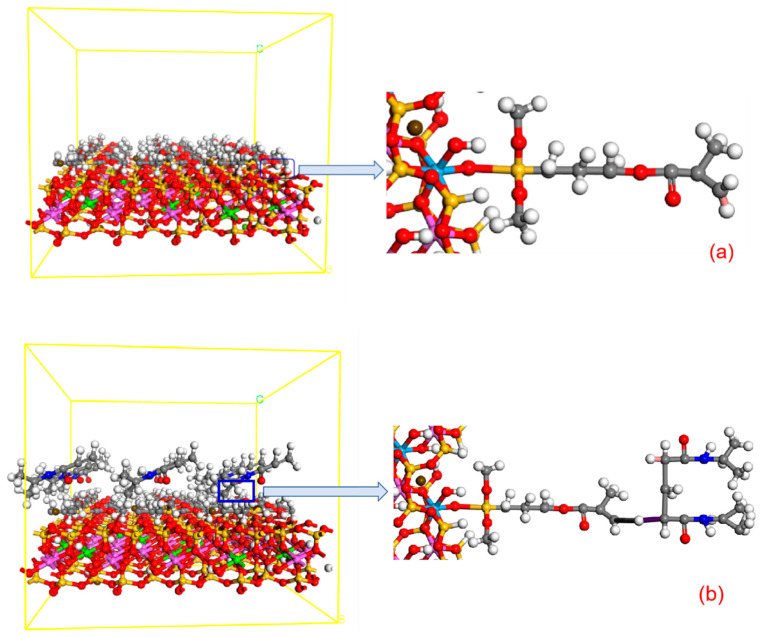
(**a**) Schematic diagram of the structure of bentonite (CMC-B-KH570) after surface modification and (**b**) Schematic diagram of the structure of temperature-sensitive modified bentonite (CMC-B-NIAPAM).

**Table 1 molecules-28-03839-t001:** Test data of transmittance and absorbance of bentonite at different temperatures.

Temperature (°C)	20	25	30	35	40	45	50	55	60	65	70
Transmittance	0.2	0.2	0.2	0.2	0.2	0.2	0.2	0.2	0.2	0.2	0.2
Absorbance	2.6	2.6	2.6	2.6	2.6	2.6	2.6	2.6	2.6	2.6	2.6

**Table 2 molecules-28-03839-t002:** Absorbance and transmittance data of modified temperature-sensitive bentonite at different temperatures.

Temperature (°C)	20	25	30	35	40	45	50	55	60	65	70
Transmittance	0.2	0.2	0.2	0.2	0.2	0.2	0.1	0.1	0.1	0.1	0.1
Absorbance	2.6	2.6	2.6	2.6	2.6	2.6	2.7	2.7	2.7	2.7	2.7

**Table 3 molecules-28-03839-t003:** Absorbance and transmittance data of modified temperature-sensitive bentonite at 45–50 °C.

Temperature (°C)	45	46	47	48	49	50
Transmittance	0.2	0.2	0.2	0.1	0.1	0.1
Absorbance	2.6	2.6	2.6	2.7	2.7	2.7

**Table 4 molecules-28-03839-t004:** Stability experiment of intelligent modified temperature-sensitive bentonite dispersion.

Dispersion Type	$\rho_{top}$	$\rho_{bottom}$	SF	Evaluation
Bentonite	1.01	1.07	0.5144	Good stability
CMC-B-NIPAM	1.05	1.06	0.5023	Good stability

## Data Availability

Not Applicable.

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
