# Peer review of "Temperature-Sensitive Modified Bentonite Based on NIPAM for Drilling Fluid: Experimental and Molecular Simulation Studies"

_molecules, 2023, doi:10.3390/molecules28093839_

Round 1
Reviewer 1 Report
This manuscript discusses the drilling fluid modified-bentonite based on NIPAM. Paying attention to drilling fluids in the wellbores' stability is essential. In this regard, applying materials that could help stabilize the wellbore's wall is a concern in the oil and gas industries. One of the main parameters to assist a higher stability rate is temperature. What is seen here is the authors have applied a range of 40 to 70 degrees Celsius, resulting in the thickening behavior of the used drilling fluid to the wellbore. This application can be more beneficial for those ranges than other drilling fluids. However, some wellbores are conditioned lower than that range; otherwise, a lower degree of temperature may be further helpful to the area with different ground and sub-surface conditions.
The introduction section is informative by adding some recent publications, especially in oil-well lightweight cement slurry in low-temperature conditions. Tables 1 and 2 range from 20 to 70 degrees Celsius; however, the results indicated in the abstract range from 40 to 70 degrees Celsius!
From Methods, some statements regarding XRD and FTIR analyses are general and must be polished in the revision stage. As a suggestion, two sub-sections of 2.3.1 and 2.3.2 can be combined by avoiding redundant statements.
Apply θ (dial reading) instead of R for equations 1 to 4. Provide a proper reference (or references) for those equations.
Figure 2(a) cannot differentiate which is transmitted and which is absorbance! This is true for Figure 4(a) too.
Provide a valid reference for the SF value, indicating which value falls within good stability.
Line 629 [Figure 3.22], it seems the authors have not converted the list of figures from their report/dissertation to an article format!
Conclusions cannot be listed and included in the digits from the results. Revise it thoroughly.
Minor editing of English language required.
Author Response
Dear editors and reviewers:
Thank you for your letter and for your comments concerning our manuscript entitled “Preparation and drilling fluid properties of temperature-sensitive modified bentonite based on NIPAM: experimental and molecular simulation studies”(ID:molecules-2362828). Those comments are all valuable and very helpful for revising and improving our paper, as well as the important guiding significance to our researches. We have read the comments carefully and have made corrections. We hope will meet with approval. Revised portion are marked in red in the paper. The main corrections in the paper and the responds to the reviewer’s comments are as the content conveyed in the document.

Reviewer 2 Report
Kindly follow the comments as per highlights.

Double check your English.
Author Response

(The authors gave the same response as above.)

Reviewer 3 Report
Dear Authors,
my comments are attached.
Best regards

Author Response

(The authors gave the same response as above.)
